# Combined Bulked Segregant Analysis-Sequencing and Transcriptome Analysis to Identify Candidate Genes Associated with Cold Stress in *Brassica napus* L

**DOI:** 10.3390/ijms26031148

**Published:** 2025-01-28

**Authors:** Jiayi Jiang, Rihui Li, Kaixuan Wang, Yifeng Xu, Hejun Lu, Dongqing Zhang

**Affiliations:** 1Xianghu Laboratory, Hangzhou 311231, China; jjy326@sina.cn (J.J.); lirihui@xhlab.ac.cn (R.L.); wangkaixuan@xhlab.ac.cn (K.W.); 2College of Life Sciences, Nanjing Agricultural University, Nanjing 210095, China; yifeng@njau.edu.cn

**Keywords:** *Brassica napus*, cold resistance, BSA-seq, RNA-seq, time-course

## Abstract

Cold tolerance in rapeseed is closely related to its growth, yield, and geographical distribution. However, the mechanisms underlying cold resistance in rapeseed remain unclear. This study aimed to explore cold resistance genes and provide new insights into the molecular mechanisms of cold resistance in rapeseed. Rapeseed *M98* (cold-sensitive line) and *D1* (cold-tolerant line) were used as parental lines. In their F_2_ population, 30 seedlings with the lowest cold damage levels and 30 with the highest cold damage levels were selected to construct cold-tolerant and cold-sensitive pools, respectively. The two pools and parental lines were analyzed using bulk segregant sequencing (BSA-seq). The G’-value analysis indicated a single peak on Chromosome C09 as the candidate interval, which had a 2.59 Mb segment with 69 candidate genes. Combined time-course and weighted gene co-expression network analyses were performed at seven time points to reveal the genetic basis of the two-parent response to low temperatures. Twelve differentially expressed genes primarily involved in plant cold resistance were identified. Combined BSA-seq and transcriptome analysis revealed *BnaC09G0354200ZS*, *BnaC09G0353200ZS*, and *BnaC09G0356600ZS* as the candidate genes. Quantitative real-time PCR validation of the candidate genes was consistent with RNA-seq. This study facilitates the exploration of cold tolerance mechanisms in rapeseed.

## 1. Introduction

Rapeseed is a significant oil crop in the cruciferous family. Its production is limited by various biological and abiotic stressors. Low temperatures are among the primary abiotic factors limiting its geographical distribution, growth, development, and productivity [1]. Low-temperature stress inhibits enzyme activities, damages membrane systems, and disrupts cell structures in plants, affecting photosynthesis, cell osmotic pressure, sugar metabolism, amino acid metabolism, and peroxide balance and resulting in varying degrees of harm [2,3,4]. However, plants have evolved a sophisticated regulatory mechanism known as cold acclimation, wherein exposure to low but non-freezing temperatures enhances their resistance to freezing, facilitating adaptation to cold stress [5,6]. Cold acclimation is a complex process, through which plants sense low environmental temperatures and modulate the expression of cold-related genes via signaling events, enhancing their cold tolerance. 

Cold sensing is the first step in plant response to cold stress and is crucial in the entire cold response process. The primary cold-sensing pathways identified to date include the membrane fluidity hypothesis, Ca^2+^ channels, the rice cold sensor COLD1/RGA1 complex, temperature sensing by phyB, and the circadian clock factor temperature sensing pathway [7,8]. In addition, potential cold sensors in plants, including cyclic nucleotide-gated channels (CNGCs) and glutamate receptors (GLRs), have been reported [7]. 

The ability of plants to acclimatize to cold conditions is a polygenic trait involving numerous gene interactions, with the c-repeat/dre binding factor (CBF) transcription factors playing a central role. The ICE1–CBF–COR signaling cascade is the most significant low-temperature signaling pathway in this process. Under low-temperature stress conditions, the transcription factor inducer of CBF expression 1 (ICE1) binds to the promoters of the CBF genes and activates their expression. Activated *CBFs* enhance downstream *cold-regulated* (*COR*) expression, improving cold resistance [9]. *ICE1* ubiquitination, ubiquitin-like modifications, and phosphorylation regulate its expression, significantly influencing the activity of its protein. At low temperatures, ICE1 is ubiquitinated by high expression of osmotically responsive 1, inducing its degradation [10], which weakens or eliminates the induction effect of ICE1 on downstream CBF-COR genes. Conversely, at low temperatures, the SIZ1 SUMO E3 ligase acts on the K393 amino acid of the ICE1 protein, performing ubiquitin-like modifications. This positively regulates ICE1 activity and reduces its degradation [11]. Under cold induction, ICE1 is phosphorylated by open stomata 1, a protein kinase involved in abscisic acid (ABA) signaling and also activated by cold stress, positively regulating plant cold tolerance; this phosphorylation interferes with HOS1 ubiquitination of ICE1, enhancing ICE1 stability under low temperatures [12]. Cold signals activate the MKK4/5–MPK3/6 cascade, which regulates ICE1 stability by phosphorylating it, weakening CBF signaling [13]. In addition to the ABA signaling pathway that integrates the ICE1–CBF–COR transcriptional cascade, other plant hormones, such as jasmonic acid, gibberellin, ethylene, and brassinolide, interact with some components of the ICE1–CBF–COR regulatory cascade in response to cold stress. Jasmonate-zim-domain protein 1/4 physically interact with ICE1/2 and inhibit their transcriptional activities, negatively regulating CBF expression and cold tolerance in *Arabidopsis* [14]. MYC2 participates in MeJa-induced cold resistance through coordinated physical interactions with ICE1 in *Musa acuminata* [15]. Achard et al. found that DELLA proteins are components of the CBF1-mediated cold stress response [16]. The key transcription factors ethylene insensitive 3 (EIN3) and EIN3-like 1 (EIL1) in the ethylene signaling pathway negatively regulate the expression of *CBFs* by binding to their promoters, modulating freezing tolerance [17]. Three transcription factors downstream of BIN2, brassinazole-resistant 1, BRI1-EMS-SUPPRESSOR 1, and CESTA, positively regulate *CBF* expression and enhance plant frost resistance [18,19]. Similarly, mitogen-activated protein kinase (MAPK) cascades have been proposed to function in cold responses. A MAPK cascade contains three protein kinases, namely, MAP kinase kinase kinase (MAPKKK, MAP3K, or MEKK), MAP kinase kinase (MAP2K, MKK, or MEK), and MAP kinase (MAPK) [7]. The MEKK1–MKK2–MPK4 cascade positively regulates cold stress in *Arabidopsis* [20]. MAPK activity is also regulated by CRLK1, inducing CBF regulators and frost resistance [21]; MKK6 activates MAPK3 and MAPK6 to enhance cold resistance in rice [22]. Additionally, several genes in pathways, such as Ca^2+^ signaling, calcium-binding proteins, reactive oxygen species (ROS) signaling, circadian rhythms, and light signaling, participate in plant cold stress responses. These include *calmodulin* (*CaM*), *CaM-like proteins*, *calcineurin B-like proteins*, Ca^2+^-dependent protein kinases, *ZAT12*, *xyloglucan endotransglucosylase/hydrolase 21*, *ascorbate peroxidase*, *circadian clock-associated 1*, *late elongated hypocotyl*, *proline/serine-rich protein*, *LUX*, and *phytochrome interacting factor3*/*4*/*7* [7,23,24,25].

There are many genes related to cold tolerance, and a variety of substances associated with cold resistance as well, such as soluble sugars (such as glucose, sucrose, fructose, and galactose), soluble proteins, membrane phospholipids, free amino acids (particularly proline), hormones (salicylic acid, abscisic acid, gibberellic acid, ethylene, cytokinin, and jasmonic acid), amines (polyamine and spermidine), glutathione, glucosinolates, carotenoids, and unsaturated fatty acids [26,27,28,29,30,31]. These substances protect cells from direct damage caused by low temperatures and help maintain the stability of cellular functions and structures.

Cold resistance mechanisms in *Arabidopsis* and rice are relatively well understood; however, they are partially elucidated in rapeseed. *BnCBF5* (*BnaC03g71900D*) and *BnCBF17* (*BnaA08g30910D*) overexpression in Westar significantly induced the expression of *BN115*, *BN28,* and *BN47* (orthologs of *Arabidopsis COR15*, *COR6.6,* and *COR47*, respectively), markedly enhanced constitutive freezing resistance, and partially regulated chloroplast development to increase the photochemical efficiency and photosynthetic capacity of transgenic plants [32,33]. *BnCOR25* overexpression in *Arabidopsis* enhances its cold tolerance [34]. Although there has been no functional research on the ICE1 genes in *Brassica napus*, ICE1 in cruciferous plants has been confirmed to regulate cold resistance in plants. *BrICE1* and *BrICE2* overexpression in *Arabidopsis* enhances ROS scavenging ability and increases cold tolerance [35]. The overexpression of *BcICE1* in rice or tobacco increases plant freezing tolerance by upregulating *CBF* [36,37]. BrrICE1.1 could bind the promoter of *CBFs* to participate in the freezing tolerance of turnip, as revealed by transcriptomics and targeted metabolomics [38]. In addition to the ICE1–CBF–COR signaling cascade pathway, other signaling pathways have been identified in plant tolerance to low temperatures. *Brassica napus cycling dof factor 1* overexpression increases cold tolerance in transgenic *Arabidopsis* by increasing *CBF1*, *COR15A,* and *COR47* expression levels [39]. *Brassica napus* calcineurin b-like protein 1 negatively regulates cold tolerance in transgenic *Arabidopsis* by weakening Ca^2+^ influx regulation [40]. *Brassica napus light-harvesting chlorophyll a/b-binding proteins 3.4* overexpression in *Arabidopsis* significantly enhances its freezing tolerance, a phenomenon possibly involving the ABA signaling pathway [41]. Overexpressed *Brassica napus tropinone reductase 1* in *Arabidopsis* increases the alkaloid content, increasing its cold resistance [42]. 

Bulked Segregant Analysis (BSA) is a common high-throughput mixed sequencing in various studies for screening candidate genes in plants. For example, Klein et al. used BSA-seq to identify *narrow odd dwarf* (*nod*) as an enhancer of *teosinte branched1* (*tb1*) in maize [43]. Xu et al. fast-mapped a Chlorophyll b synthesis-deficiency gene in barley via BSA-seq [44]. In addition, Guo et al. identified candidate genes controlling chilling tolerance in rice using BSA-seq and RNA-seq [45]. Ye et al. identified candidate genes associated with the albino phenotype in *Brassica napus* through BSA-seq and RNA-seq [46]. Yan et al. identified chilling-tolerant genes in maize via BSA-seq [47]. Similarly, Zhang et al. mapped the waxy gene in *Brassica napus* using BSA-seq [48]. Also, de novo transcriptome or genome assembly can better identify key genes and pathways associated with abiotic stress in *Brassica* [49,50].

In this study, rapeseed *M98* (cold-sensitive line) and *D1* (cold-tolerant line) were used as parental lines. BSA-seq analysis was conducted on the parental lines as well as the cold-tolerant and cold-sensitive pools from their F2 segregating population. In addition, rapeseed *M98* and *D1* were subjected to low-temperature treatment with a gradual and continuous decrease in temperature for RNA-seq analysis. Finally, combined BSA-seq and RNA-seq revealed three candidate genes for cold resistance. This study aimed to explore cold resistance genes in rapeseed *D1* and provide new insights into the molecular mechanisms of cold resistance in rapeseed.

## 2. Results

### 2.1. Phenotypic Evaluation of D1, M98, and Their F_2_ Progeny Under Cold Stress

The phenotypes of the two inbred rapeseed lines, *D1* and *M98*, were assessed under cold stress. There were no evident phenotypic differences between the two lines before or during cold treatment (Figure 1a,b). Rapeseed *D1* was determined to be a cold-tolerant line that exhibited a 75% survival rate (SR), with healthy leaves after cold stress (Figure 1c). In contrast, rapeseed *M98* was a cold-sensitive line, with all seedlings dying, seedlings and leaves wilting (Figure 1c). 

A total of 1,433 individual plants from the F_2_ generation were cultivated. Phenotypic identification revealed 125, 629, 227, 371, and 88 rapeseed seedlings with cold damage levels of 0, 1, 2, 3, and 4, respectively (Figure 1d–h). Resistance distribution was approximately normal, indicating that the cold resistance of rapeseed was consistent with the inheritance of the quantitative characteristics. Thirty rapeseed samples with cold damage level 0 and 30 with cold damage level 4 were selected as cold-tolerant and -sensitive pools, respectively. The phenotypes of the cold-tolerant and -sensitive plants are shown in Figure 1d,e. 

### 2.2. BSA-seq Data Analysis

The two parental and two progeny pools were re-sequenced using the Illumina HiSeq. The pools produced 97.58 Gb of clean data with a Q30 percentage of 95.73%. The average alignment efficiency of the samples to the reference genome was 98.17%, with an average coverage depth of 41.29× and a genome coverage of 95.08% (with at least one base covered). The AT and CG bases in the samples showed minimal segregation, and the fragment size distribution for each insertion displayed a unimodal normal distribution, indicating that the sequencing data quality was acceptable. The alignment efficiency to the rapeseed reference genome was high, making it suitable for subsequent analyses.

After mapping with the “ZS11.v0” reference genome, these four re-sequence samples generated 6,564,931 SNPs and 1,575,516 Indels. Through qtlr, the target intervals were intercepted based on the G’-value (Figure 2). The results showed that seven peak values were significant; however, except for a single peak on Chromosome C09, the remaining six peaks were clustered on Chromosome C02. Among these seven intervals, the largest was 12.53 Mb, and the smallest was 1.72 Mb (Appendix A). 

### 2.3. Identification of Candidate Genes in Candidate Interval

The BSA-seq results showed that seven peak values were significant, but except for a single peak located on chromosome C09, the remaining six peaks were clustered on chromosome C02. Moreover, the six peaks on chromosome C02 were not only closely clustered but also covered more than half the chromosome, suggesting that this may be due to the background differences between the parents. So, to effectively identify the major genes associated with cold stress in the cold-resistant materials, we focused on the interval on chromosome C09, which was considered the candidate interval (CI) for further analysis. In the CI, 85 genes had variants in their exons, 69 of which were nonsynonymous (Appendix A). According to statistical analysis, most genes clustered with mutations in the SNPs. For example, 21 genes had nonsynonymous variants only, 20 had synonymous and nonsynonymous variants, and 16 had synonymous mutations only. The remaining genes had SNP and Indel variants to varying extents (Figure 3).

Functional prediction of the 69 candidate genes in CI was performed. Enrichment analysis of Gene Ontology (GO) annotations and Kyoto Encyclopedia of Genes and Genome (KEGG) pathways showed the processes associated with cold response, such as the glycerolipid biosynthetic process (GO:0045017), glycerolipid metabolic process (GO:0046486), lipid biosynthetic process (GO:0008610), oxidoreductase activity acting on single donors with the incorporation of molecular oxygen and incorporation of two oxygen atoms (GO:0016702 and GO:0016701), and mitogen-activated protein kinase (MAPK) signaling pathway-plant (bna04016) (Appendix A). 

### 2.4. RNA-seq Data Analysis

To reveal the relationship between the increased cold resistance in rapeseed *D1* and changes in gene expression, leaves subjected to low-temperature treatment were used for transcriptome sequencing. After filtering low-quality reads, the contents of clean reads, Q30, and GC in each sample were not less than 6.04 G, 97.02%, and 45.56%, respectively (Appendix A). Through alignment analysis, 88.91–91.76% of the clean data was mapped to the genome of *Brassica napus* (ZS11.v0), which implied the reliable quality of the RNA-seq (Appendix A). Principal component analysis (PCA) showed that the biological replicates of the two rapeseed varieties before and after low-temperature treatment were consistent (Appendix A). 

Low-temperature response is a dynamic and complex process that usually changes the expression pattern of many genes [51]. In the cold-sensitive rapeseed *M98*, many genes were not expressed after low-temperature treatment; however, cold-tolerant rapeseed *D1* exhibited no significant alterations in the number of expressed genes, except for the 6th sampling time point (−4 °C, 5 h) (Figure 4a). Under the thresholds of absolute log_2_(fold-change) ≥ 1 and padj ≤ 0.05, the numbers of DEGs at seven time points in rapeseed *D1* and *M98* were almost all more than 10,000, and the most were 17,694. The numbers of DEGs in *D1* vs. *M98* at seven time points showed only a slight difference, all approximately 5000 (Figure 4b). This may be owing to the characteristics of the different varieties that respond to low-temperature treatments. For the analysis of DEGs between two adjacent time points, the number of DEGs in *D1* and *M98* showed opposing trends (log_2_(fold-change) ≥ 1 and padj ≤ 0.05); the number of DEGs tended to increase in *D1* and decrease in *M98* (Figure 4c). The number of DEGs in *D1* between two adjacent time points showed a marked increase between time points 5 and 6. In contrast, there were few DEGs between the 5th and 6th time points in *M98*. These results indicate that the gene expression changes during this period were crucial, and there was a high possibility that the DEGs contained key factors that determined the differences in cold resistance between the two parental lines.

The GO analysis of the DEGs of the 6th vs. 5th time points in *D1* showed that many enriched terms were significantly correlated with cold resistance, such as sugar metabolism (GO:0005976, GO:0006073, GO:0044042, GO:0044264, and GO:0044262), amino acid catabolism (GO:0009074 and GO:0006569), amine catabolism (GO:0009310), intracellular or extracellular skeleton and composition (GO:0005618, GO:0030312, GO:0048046, and GO:0005576), glucosyltransferase activity (GO:0046527 and GO:0035251), microtubule/tubulin binding or microtubule motor activity (GO:0008017, GO:0015631, and GO:0003777), and cytoskeletal protein binding (GO:0008092) (Figure 5a). KEGG pathway analysis of the gene set of the 6th vs. 5th time points in *D1* revealed that many enriched pathways were significantly correlated with cold resistance, such as photosynthesis-antenna proteins (bna00196), glucosinolate and carotenoid biosynthesis (bna00966 and bna00906), amino acid metabolism and degradation (bna00270, bna00280, bna00380, bna00330, bna00260, bna00350, bna00360, bna00310, bna00340, and bna00400), fatty acid metabolism and degradation (bna00071 and bna00592), MAPK signaling pathway-plant (bna04016), and glutathione metabolism (bna00480) (Figure 5d). These results indicate that between these two time points, many regulatory pathways related to cold resistance were involved in the *D1* response to low-temperature stress. In addition, GO term and KEGG pathway analyses at the D5 and D6 intersection were performed. GO indicated that many enriched terms were significantly correlated with cold resistance, such as photosynthesis (GO:0015979), fatty acid biosynthesis and metabolism (GO:0006633, GO:0006631, GO:0008610, and GO:0044255), photosystem (GO:0009522, GO:0009521, GO:0034357, GO:0009538, GO:0009523, and GO:0009654), cell structure (GO:0044436, GO:0009579, GO:0005576, and GO:0005618), and catalase activity (GO:0004096) (Figure 5b). KEGG showed that many enriched pathways were significantly correlated with cold resistance, such as photosynthesis (bna00196 and bna00195); fatty acid biosynthesis, elongation, and metabolism (bna00062, bna00561, bna01040, bna00592, bna00591, bna00071, and bna01212); Circadian rhythm-plant (bna04712); anomic acid metabolism (bna00380, bna00330, bna00310, and bna00360); peroxisome (bna04146); thiamine metabolism (bna00730); and MAPK signaling pathway-plant (bna04016) (Figure 5e). These results also imply that many regulatory pathways related to cold resistance were continuously involved in the *D1* response to low-temperature stress at both time points. The above genes in the 6th vs. 5th time points and the intersection of the 5th and 6th time points in *D1* were combined into the gene set CR1 for further analysis (Appendix A).

The GO analysis of the gene set of the 6th vs. 5th time points in *M98* showed that some enriched terms were significantly correlated with cold resistance, such as sugar metabolic process (GO:0006073, GO:0044042, GO:0044264, GO:0005976, and GO:0046527) and peroxisome (GO:0005777, GO:0005778, GO:0005779, GO:0031231, and GO:0044439) (Figure 5c). KEGG pathway analysis of the gene set of the 6th vs. 5th time points in *M98* showed no significantly enriched pathways. In addition, GO term and KEGG pathway analyses of the intersections of the 5th and 6th time points in *M98* were performed. GO indicated that no enriched terms were significantly correlated with cold resistance (Figure 5e), while the KEGG pathway analysis showed many significantly enriched pathways, such as photosynthesis (bna00195), amino acid metabolism or degradation (bna00310, bna00380, bna00280, and bna00330), unsaturated fatty acid biosynthesis (bna01040), and fatty acid degradation (bna00071) (Figure 5f). The above genes in the 6th vs. 5th time points and in the intersection of the 5th and 6th time points in *M98* were combined into gene set CR2 for further analysis (Appendix A).

### 2.5. Time-Course Differential Expression Analysis Between D1 and M98

MaSigPro analysis was used to investigate the diverse expression patterns at the seven time points in *D1* and *M98*. We identified 1851 genes with the same change trend at three adjacent time points as time-course DEGs (|log2fc| ≥ 1, padj ≤ 0.05) in *D1* vs. *M98*. Nine clusters corresponding to differential expression patterns were identified using the Hierarchical clustering (hclust) method (Figure 6, Appendix A).

The dynamic expression pattern was divided into two dominant expression types using the time-course clustering analysis. The first type was the dominant expression pattern of *D1* (N = 930) and was represented by Clusters 1, 2, 5, 6, and 9. The second type was *M98* (N = 919), which was represented by Clusters 3, 4, 7, and 8 (Figure 6). To further elucidate the potential functions, GO enrichment analysis was performed for each cluster. The significantly different and top-ranked GO terms are shown in Figure 6. Some DEGs were enriched in DNA integration (GO:0015074) and peptidase activity terms (GO:0004866, GO:0030414, and GO:0061134) related to gene expression regulation and protein metabolism, while others were enriched in fatty acid biosynthesis (GO:0006633) and photosystem terms (GO:0009538 and GO:0009523) related to cold resistance in Cluster 1, which represented a group of genes that were downregulated in *D1* but almost not expressed in *M98* (Figure 7). The GO terms of Cluster 2 were closely related to cell structures, such as carbohydrate biosynthesis or metabolism (GO:0016051 and GO:0044262) and intracellular and extracellular structures (GO:0005618 and GO:0030312) (Figure 7a). In Cluster 5, the GO terms were enriched in carbohydrate biosynthesis, metabolic or catabolic processes (GO:0043648, GO:0016052, and GO:0034637), cellular amino acid metabolic processes (GO:0006520), and serine peptidase or hydrolase activity (GO:0008236 and GO:0017171) (Figure 7a). The genes in Cluster 6 were predominantly upregulated in *D1*, and the enrichment terms were related to gene expression regulation; amino acid metabolism; basic physiological processes, such as rRNA processing (GO:0006364 and GO:0016072); and arginine biosynthetic or metabolic processes (GO:0006525 and GO:0006526) (Figure 7a). The enriched GO terms in cluster 9 participate in the modification of DNA, RNA, proteins, or lipids, such as S-adenosylmethionine-dependent methyltransferase activity (GO:0008757) (Figure 7a). The vitamin biosynthetic or metabolic processes (GO:0006766, GO:0006767, and GO:0009110), nuclear export signal receptor activity (GO:0005049), and nucleocytoplasmic carrier activity (GO:0140142) were enriched in Cluster 3 (Figure 7b). Notably, the enriched GO terms in Cluster 4 were related to translation or post-transcriptional regulation of gene expression (GO:0006417, GO:0006448, GO:0006449, GO:0006452, and GO:0010608) (Figure 7b). In Cluster 7, the enriched terms were related to photosynthesis and photosystems (GO:0019684, GO:0009538, GO:0009522, and GO:0009654) (Figure 7b). The enriched GO terms of Cluster 8 included phosphorelay or intracellular signal transduction (GO:0000160 and GO:0035556) and O-methyltransferase activity (GO:0008171) (Figure 7b).

### 2.6. Co-expression Network Construction and Overlap Analysis with MaSigPro

Based on the result of MaSigPro analysis, the genes predominantly expressed in *D1* (N = 930) and *M98* (N = 919) (Figure 6) were used to construct co-expression networks through weighted gene co-expression network analysis (WGCNA). This was done to explore the correlation between the modules and gene expression (Appendix A). Following the removal of the gray module considered invalid in *D1*, eight modules were identified, namely, brown (113), turquoise (316), yellow (107), black (43), blue (114), pink (36), green (81), and red (51) (Appendix A). By removing the gray module considered invalid in *M98*, nine modules were identified, namely, magenta (36), red (61), blue (164), pink (46), black (57), yellow (86), turquoise (171), brown (152), and green (84) (Appendix A). In addition, in the co-expression network of *D1*, the brown module was significantly correlated with the 5th time point (-4 °C, 1 h) (Pearson’s r = 0.81), pink was correlated with the 6th time point (−4 °C, 5 h) (Pearson’s r = 0.81), and red was correlated with the 7th time point (−4 °C, 24 h) (Pearson’s r = 0.79) (Figure 8a). In the co-expression network of *M98*, the red module was significantly correlated with the 7th time point (−4 °C, 24 h) (Pearson’s r = 0.98), pink was correlated with the 4th time point (−2 °C, 1 h) (Pearson’s r = 0.97), yellow was correlated with the 1st time point (4 °C, 3 days) (Pearson’s r = 0.94), magenta was correlated with the 6th time point (−4 °C, 5 h) (Pearson’s r = 0.88), and turquoise was correlated with the 2nd time point (2 °C, 1 h) (Pearson’s r = 0.79) (Figure 8b). 

Overlapping analysis of the genes in the five clusters from MaSigPro and eight modules from WGCNA was performed on *D1*, whereas gene overlapping analyses in four clusters and nine modules were performed on *M98*. The results showed that each module overlapped with a specific cluster (Figure 8c,d). Genes that appeared simultaneously in clusters and modules, which were combined into the gene set CR3 for further analysis (Appendix A), were considered crucial in the biological function of cold resistance. The top three percentiles were selected to form the following new modules: D1-cluster2-turquoise, D1-cluster9-green, D1-cluster1-yellow, M98-cluster8-red, M98-cluster3-turquoise, and M98-cluster7-pink (Appendix A). 

The GO enrichment analysis of the six new modules was performed. In the D1-cluster2-turquoise, GO was enriched in carbohydrate biosynthesis and metabolic processes (GO:0034637, GO:0016051, GO:0044262, GO:0030243, and GO:0030244) (Appendix A). The GO terms of D1-cluster9-green were closely related to phospholipase or lipase activity (GO:0004435, GO:0004629, GO:0004620, and GO:0016298) (Appendix A). Notably, the enriched GO terms involved in photosystem or photosynthesis (GO:0015979, GO:0009538, GO:0009522, GO:0009521, GO:0034357, GO:0044436, and GO:0009579) belonged to D1-cluster1-yellow (Appendix A). Furthermore, the enriched GO terms in M98-cluster8-red were tryptophan or amine catabolic processes (GO:0006569, GO:0009310, GO:0019441, GO:0042402, and GO:0009074) (Appendix A). No terms were significantly enriched in M98-cluster3-turquoise. Finally, the only enriched term was chromatin binding (GO:0003682) in M98-cluster7-pink (Appendix A). These results show that the new modules were enriched in GO terms related to cold resistance.

### 2.7. Identification of RNA-seq Candidate Genes

Based on the eigengene connectivity values, the top five genes with the highest kME values were selected as hub genes in each new module (Appendix A). However, hub genes do not fully represent the biological functions of the modules; therefore, considering the functions of other genes in the co-expression network is essential. To visually observe the genes in each module, the interaction network of high-weight gene pairs was visualized using Cytoscape [52] (Appendix A).

In the D1-cluster2-turquoise, only *cytochrome b5 isoform c* (*BnaC04G0611700ZS*), which modulates fatty acid desaturation [53], was significantly associated with cold resistance (Appendix A, Appendix A), whereas there were six genes that were closely related to plant cold resistance in D1-cluster9-green (Appendix A, Appendix A), including *CBF2* (*BnaA08G0172700ZS*), *COR413-plasma membrane 2* (*COR413PM2)* (*BnaC07G0389700ZS*), *dehydration-responsive element-binding protein 2A* (*DREB2A)* (*BnaC02G0020400ZS* and *BnaC02G0020500ZS*), *calmodulin-like 24* (*CML24)* (*BnaA04G0130900ZS*), and *early responsive to dehydration 7* (*ERD7)* (*BnaA07G0024700ZS*) [6,54,55,56,57]. *Photosystem I subunit H 1* (*PSAH-1)* (*BnaC01G0424400ZS*), which encodes the subunit H of the Photosystem I reaction center subunit VI and may contribute to the response to cold, was identified in D1-cluster1-yellow (Appendix A, Appendix A). In addition, *COR28* (*BnaA08G0138800ZS*), which encodes night-time repressors that integrate *Arabidopsis* circadian clock and cold response [58] and *cold acclimation protein 160* (*CAP160*) (*BnaA03G0487600ZS*), which encodes a low-temperature-induced protein [59], were identified in M98-cluster8-red (Appendix A, Appendix A). Finally, *filamenting temperature-sensitive Z 2-1* (*FTSZ2-1)* (*BnaC04G0548700ZS*), which is a chloroplast division gene [60], and *gibberellin 2-oxidase 1* (*GA2OX1)* (*BnaC03G0214800ZS*), which participates in gibberellin catabolism, were found in M98-cluster3-turquoise (Appendix A, Appendix A). 

### 2.8. Identification of Candidate Genes by Combining BSA-seq and RNA-seq

To identify candidate genes related to rapeseed *D1* cold resistance, association analyses were performed using BSA-seq and RNA-seq. First, the gene sets CR1, CR2, and CR3, obtained from the previous RNA-seq analysis, intersected with the CI. Subsequently, 12, 13, and 2 genes were identified, respectively (Appendix A). Among them, *BnaC09G0352400ZS*, *BnaC09G0353200ZS*, and *BnaC09G0356600ZS*, which are related to cold response, were identified as candidate genes (Table 1). 

In addition, the DEGs of D1_T vs. M98_T at the seven time points intersected with CI; however, no intersections were observed between them (Appendix A). This could be because the 1st time point was 3 days after cold acclimation at 4 °C, where gene expression was less intense than that at the beginning of cold-temperature treatment. Therefore, the DEGs of D1_T vs. M98_T at the other six time points intersected with CI. Only one gene, *BnaC09G0354200ZS*, was identified in this study (Appendix A). The CI was intersected with the DEGs in the chilling (2nd and 3rd time points, T ≥ 0 °C) and freezing (4th–7th time points, T ≤ 0 °C) treatment groups. The results showed that two genes (*BnaC09G0354200ZS* and *BnaC09G0350200ZS*) and one gene (*BnaC09G0354200ZS*) were identified in the chilling and freezing groups, respectively (Appendix A). However, BnaC09G0350200ZS is a LIM domain protein involved in mitotic DNA replication checkpoint signaling and responses to ionizing radiation. It is not related to cold resistance and should be eliminated from the list of candidate genes. 

To sum up, *BnaC09G0354200ZS*, *BnaC09G0353200ZS*, and *BnaC09G0356600ZS* were identified as candidate genes. Nucleic acid and protein sequences are shown in Appendix A. The three candidate genes had identical nucleotide and protein sequences in *M98* to those of ZS11.v0. The *BnaC09G0354200ZS* gene had three SNPs in the coding region, two of which were nonsynonymous mutations; one SNP at position 146 (c.146T > C) caused the nonsynonymous mutation p.Ile49Thr, which replaces isoleucine with threonine in the protein sequence, and another SNP at position 325 (c.325G > A) caused the nonsynonymous mutation p.Ala109Thr, which replaces alanine with threonine in the protein sequence. *BnaC09G0353200ZS* has two SNPs in the coding region, one of which is a nonsynonymous mutation, and one SNP at position 299 (c.224A > C) causes the nonsynonymous mutation p.Asp100Ala, which replaces aspartic acid with alanine in the protein sequence. *BnaC09G0356600ZS* has only one SNP in the coding region, which is a nonsynonymous mutation; one SNP at position 224 (c.224C > A) causes the nonsynonymous mutation p.Ala75Asp, which replaces alanine with aspartate in the protein sequence. 

### 2.9. qRT-PCR Analysis of Candidate Genes

Three candidate genes from the combination *of BSA-seq* and *RNA-seq* and five DEGs from RNA-seq were selected for validation using quantitative real-time PCR (qRT-PCR). The results showed that the relative expression levels of the DEGs determined through qRT-PCR corresponded well with the RNA-Seq data (Appendix A), verifying the reliability of the transcriptome data.

In addition, the expression patterns of the three candidate genes before and after cold treatment could be divided into two groups. One group includes only one gene *BnaC09G0354200ZS*, which shows hardly a change in *D1* but increased expression in *M98* after cold stress. The other group consists of two genes *BnaC09G0353200ZS* and *BnaC09G0356600ZS*, which exhibit the similar expression patten (either increased or decreased after cold stress) in both *D1* and *M98* (Appendix A). Therefore, it is inferred that BnaC09G0354200ZS in the first group may be a specific negative regulatory cold resistance gene in *D1* and *M98*, while the genes in the second group may be common to cold stress response in rapeseed.

## 3. Discussion

Low-temperature stress includes chilling (0–15 °C) and freezing (<0 °C) stresses. Chilling stress hardens membranes, destabilizes protein complexes, and impairs photosynthesis, inhibiting plant growth, while freezing stress induces intracellular and extracellular ice crystal production, causing mechanical damage and plant death [77,78].

Rapeseed is the only overwinter oil crop; however, its growth and development are susceptible to cold stress, resulting in yield loss [79,80]. “Rice-rapeseed rotation” leads to tight intervals between them, delayed planting of rapeseed, and low temperatures for the growth of rapeseed seedlings. Extreme temperatures are becoming more frequent owing to global climate change, and low temperatures pose critical threats to rapeseed growth in autumn, winter, and spring. In addition, studies of the mechanisms of cold resistance in rapeseed are limited, necessitating further investigation. Hence, studying the molecular mechanisms underlying cold resistance in rapeseed, exploring rapeseed cold-tolerance-related genes, and cultivating cold-resistant rapeseed varieties are crucial to stabilizing rapeseed yield.

Mapping of candidate cold tolerance genes using BSA-seq has been performed in several species, including maize, rice, potato, and cucumber [47,81,82,83,84]. The combination of BSA-seq and RNA-seq to identify genes with unclear cold tolerance functions or regulatory mechanisms and exploring the potential relationships among them will benefit the understanding of the regulatory networks of cold-tolerant genes in rapeseed and the creation of cold-resistant rapeseed germplasm. In this study, the cold-tolerant line *D1* and cold-sensitive line *M98* were used to identify the unknown crucial genes related to cold tolerance at the seedling stage, using BSA-seq and RNA-seq combined strategies.

Seven intervals of cold tolerance were identified through a G’-value analysis. Among them, six intervals were clustered on Chromosome C02 and distributed in more than half of the chromosome, which was speculated to be caused by background differences between parents. Therefore, we speculated that a single interval on Chromosome C09 plays a major role in cold resistance (Figure 2). Thus, to identify the major genes responsible for cold resistance, we focused on the interval on Chromosome C09 in this study. The results of GO and KEGG analyses showed that the genes in the single interval on Chromosome C09 were indeed related to cold tolerance (Appendix A).

RNA-seq revealed that the number of DEGs of the 6th vs. 5th time points had the maximum value in *D1* and showed the maximum difference between *D1* and *M98* (Figure 4c). These results suggest that changes in gene expression during this period are critical for cold resistance in rapeseed. GO and KEGG analyses of the 6th vs. 5th time point genes and the intersecting genes of the 5th and 6th time points in *D1* and *M98* were performed. The results showed that many GO term and KEGG pathway enrichments were associated with cold resistance (Figure 5). Therefore, all genes in CR1 and CR2 intersected with all genes in CI, and three candidate genes, namely, *BnaC09G0354200ZS, BnaC09G0353200ZS*, and *BnaC09G0356600ZS*, were identified (Table 1, Appendix A). *BnaC09G0354200ZS* is a homologous gene of *Arabidopsis thaliana cytokinin response factor 3* (*CRF3)* (AT5G53290), a cytokinin response factor 3. It encodes a member of the ERF (ethylene response factor) subfamily B-5 within the ERF/AP2 transcription factor family. *AtCRF3* was rapidly and strongly induced under low temperature, but AtCRF3 did not affect the expression of the *COR* genes; however, it responded to cold via a cytokinin two-component signaling (TCS)-independent pathway [66,67]. As well, AtCRF3 is an integral part of the auxin, cytokinin, and abscisic acid pathways [85]. *AtCRF3* also is directly regulated by phytochrome interacting factor 3-like 5 to inhibit seed germination [86]. It can be concluded that CRF3 has diversified functions. Although its regulatory pathway in the plant cold resistance process remains unclear, it is clearly associated with plant cold resistance. BnaC09G0353200ZS encodes a chloroplast chaperonin-60beta2 (CPN60B2) protein, which is required for the formation of a normal chloroplast division apparatus, primarily by influencing the folding of chloroplast division-related proteins [87]. CPN60B2 play a role in preventing the thermal denaturation of Rubisco activase in *Arabidopsis* [88] and can be induced by heat stress in rapeseed [87]. Interestingly, CPN60B2 can also be dissociated from the chloroplast membrane, helping cold-resistance-related proteins on the membrane to be modified in response to cold stress in *Arabidopsis* [65]. It can be inferred that BnaC09G0353200ZS should participate in the process of plant high- or low-temperature stress. The homologous gene of *BnaC09G0356600ZS* in *Arabidopsis* is *thylakoid luminal 17* (*TL17)* (*AT5G53490*), which encodes a thylakoid lumen protein [70]. There have been no reports on the biological function of TL17 so far. However, due to its identity as a thylakoid protein, it is speculated to be involved in photosynthesis, thereby mediating the cold resistance process.

Additionally, through the overlap of time-course analysis and WGCNA, key genes with significant changes at specific time points that are highly co-expressed with other genes were identified. These genes may be crucial in various biological processes, such as cold resistance in plants. Through the hub genes and interaction networks of high-weight gene pairs, 12 DEGs were identified, most of which were classic downstream cold-resistance genes in plants, such as *CBF2*, *COR413PM2*, *DREB2A*, and *COR28*. Except for DREB2A, all others are integral components of the most significant low-temperature signaling pathway, the ICE1–CBF–COR signaling cascade. DREB2, like CBF/DREB1, is one of the largest subfamilies in the AP2/ERF transcription factor family. CBF/DREB1 proteins play a crucial role in regulating cold and frost tolerance, while DREB2 proteins are important for the regulation of drought, salt, osmotic stress, heavy metals, and extreme temperatures [89]. Moreover, the other seven candidate genes are also more or less related to the process of plant cold tolerance. *CML24* and *ERD7* expression can be induced by cold stress [56,57]. The ERD7 family can also promote cold and freezing tolerance, and the expressions of some cold-induced genes, CBF1/2/3 and COR15A/47, were downregulated in hHH mutant, having the genotype erd7^+/−^ edn2^−/−^ edn1^−/−^, and the ERD7 family also may reduce the risk of cold or freezing by mediating membrane lipid composition through membrane metabolism and/or trafficking [57]. PSAH-1 is only known to be involved in the light reactions of photosynthesis [90], and it is speculated to be related to cold tolerance. CPA160 is a low-temperature stress protein in spinach [59]. FTSZ2-1 is a chloroplast division gene [60]; it is speculated that it responds to cold stress by affecting the photosynthesis in chloroplasts. The transcript levels of GA2ox1 may be increased by cold stress through a CBF1-independent manner [16]. However, these DEGs are not located on chromosome C09. It can be inferred that these DEGs may be regulated by candidate genes to respond to cold stress. They could be downstream target genes of the candidate genes, which requires further research to confirm. Additionally, the overlapping rate of time-course analysis and WGCNA was not high in this study (Figure 8a–b). This may be because gene changes during cold tolerance are complex. Some genes exhibited similar expression trends at different time points; however, they exhibited no close relationships in the co-expression network. Therefore, to avoid missing potential cold-resistant genes, all genes in CR3 intersected with all genes in CI, and *BnaC09G0356600ZS* was identified again (Table 1 and Appendix A). 

Finally, in the more stringent overlap analysis among all DEGs of D1_T vs. M98_T between the 2nd and 7th time points and all genes in the CI group, *BnaC09G0354200ZS* was re-identified and confirmed as the only candidate gene (Appendix A). This suggests that *BnaC09G0354200ZS* plays a functional role during the low-temperature treatment process under chilling and freezing conditions. 

To sum up, the candidate genes leading to differences in cold resistance between rapeseed *D1* and *M98* were *BnaC09G0354200ZS* (*CRF3*), *BnaC09G0353200ZS* (*CPN60B2*), and *BnaC09G0356600ZS (TL17)*. According to the expression patterns of the three candidate genes before and after cold treatment, it is inferred that *BnaC09G0354200ZS* may be a specific negative regulatory cold resistance gene in *D1* and *M98*, while the genes *BnaC09G0353200ZS* and *BnaC09G0356600ZS* may be common to cold stress response in rapeseed. This seems to suggest that *BnaC09G0354200ZS* is the first candidate gene. However, AtCRF3 has been shown not to affect the expression of *COR* genes [67]. Among the twelve DEGs, only *GA2ox1* has been clearly demonstrated to be involved in cold response independent of CBF [16]. In contrast, *CBF2*, *COR413PM2*, and *COR28* are involved in the CBF–COR pathway. Moreover, ERD7 can regulate the expression of *CBF* and *COR* [57]. It is speculated that CRF3 may be associated with the CBF–COR pathway in rapeseed, or the other two candidate genes play a role in the CBF–COR pathway. CPN60B2 forms the chloroplast division apparatus by influencing the folding of chloroplast division-related proteins. It is essential for the formation of a normal chloroplast division apparatus [88]. On the other hand, the DEG *FTSZ2-1* is a chloroplast division gene [60]. Therefore, it is speculated that *FTSZ2-1* may be a downstream target of CPN60B2. Similar to the relationship of FTSZ2-1 and CPN60B2, TL17 and PSAH1 may interact with each other. LT17 is a thylakoid lumen protein [70], which is involved in photosynthesis. And PSAH1 is also involved in photosynthesis [90]. Overall, *BnaC09G0354200ZS* (*CRF3*), *BnaC09G0353200ZS* (*CPN60B2*), and *BnaC09G0356600ZS (TL17)* could all be candidate genes. Furthermore, the genetic distance of the three candidate genes is very close, only 0.5 Mb (Table 1); this suggests that they may interact with each other and jointly regulate the cold tolerance process in rapeseed. Their functions require further investigation. This study proved that the combined analyses of BSA-seq and RNA-seq provide an effective strategy to explore candidate genes and improve the understanding of the molecular mechanism of rapeseed response to cold stress.

## 4. Materials and Methods

### 4.1. Cold Tolerance Evaluation and Treatment Conditions

Rapeseed *D1* and *M98* were surface sterilized in 75% ethanol for 1 min, sterilized in 10% sodium hypochlorite for 10 min, washed three to five times with sterilized water, germinated in pots containing sterilized soil mixture (nutrient soil: vermiculite: perlite = 2:1:1), and incubated in the plant chamber at 22 °C (16/8 h, light/dark). When the plants grew three leaves, the seedlings were moved into a −4 °C plant growth chamber for cold treatment for 24 h and returned to the plant chamber at 22 °C (16/8 h, light/dark), and the survival rate (SR) after the seedling recovery was investigated. SR (%) = (Number of surviving plants after cold stress) / (Total number of plants treated).

After identification, rapeseed lines *M98* and *D1* were determined to be cold-tolerant and cold-sensitive, respectively. They were crossed on 11 March 2023. *M98* was used as the female parent, and *D1* was the male parent. Hybrid seeds were sown in a phytotron to obtain the F_2_ generation seeds via self-crossing between June and September 2023. Two parents and F_2_ seeds were planted in the field (30°37′23″ N, 120°8′45″ E) on 15 October 2023. When the plants had four leaves each, rapeseed leaves from the F_2_ population were collected for genomic DNA extraction. Rapeseed cold resistance was evaluated on 27 December 2023, after 5 days of the coldest weather on 21 December 2023 (the highest temperature was 0 °C, and the lowest was −6 °C), using the cold resistance assessment method of Quality Control Standards for Rapeseed Germplasm Resource Data [91]. According to this method, cold damage to rapeseed after 3–5 days of snowmelt or severe frost thawing was investigated. The degree of cold damage in each rapeseed plant was divided into five levels, namely, 0, 1, 2, 3, and 4. The rapeseed with cold damage level 0 showed normal and no cold damage phenotype, whereas that with cold damage level 1 showed that only a few large leaves were affected by frost damage, and the affected leaves were locally atrophic or scorched. The rapeseed with cold damage level 2 showed that half of the leaves were frozen damaged, and the damaged leaves were partially or mostly atrophic or scorched; however, the core leaves were normal. The rapeseed with cold damage level 3 showed that all the leaves were damaged by cold, the damaged leaves were partially or mostly atrophic or scorched, the core leaves were normal or slightly damaged, and the plants could resume growth. The rapeseed with cold damage level 4 showed that all leaves, including core leaves, were damaged, and the plant tended to die. The historical temperature records are shown in Appendix A.

Furthermore, rapeseed *M98* and *D1* were planted in the plant chamber at 22 °C (10/14 h, light/dark). When the plants had five leaves, half of the seedlings were moved into a 4-°C plant growth chamber for cold acclimation for 3 days, and the plant growth chamber was imposed at 2, 0, and −2 °C for 1 h and −4 °C for 24 h. The third leaves from the top were sampled at 4 °C (before the chamber temperature was changed), 2 °C (1 h), 0 °C (1 h), −2 °C (1 h), −4 °C (1 h), −4 °C (5 h), and −4 °C (24 h). Samples were collected from the control group at the same seven time points (Appendix A). All samples were quick-frozen in liquid nitrogen and stored at −80 °C.

### 4.2. Pool Construction and BSA-seq

In the F_2_ population, 30 rapeseed plants with cold damage level 0 and 30 with cold damage level 4 were identified as extremely cold-tolerant and cold-sensitive gene pools, respectively. DNA was extracted using the MolPure Plant RNA Kit (Yeasen, Shanghai, China), following the manufacturer’s protocol. The parent DNAs and two-pool DNA libraries were constructed according to standard procedures. The two pools and their parents were re-sequenced using Illumina (Novogene, Beijing, China). The whole-genome resequencing depths of the parents and two pools were 15× and 30×, respectively.

### 4.3. BSA-seq

First, reads containing adapter sequences and low-quality raw data sequences were filtered and removed using FASTP (v0.23.1). Next, the clean reads were aligned to the *Brassica napus* reference genome (ZS11.v0) using BWA (0.7.17). The SAMs were transferred to the BAM using the same samtools (v1.13). Duplicates in the alignment results were removed using the Picard software. The UnifedGenotyper module of the GATK (v4.3) software was used to detect SNPs and indel markers in multiple samples. The SNPs of two pools were used to determine BSA through the QTL-seqr package, and the G’-value of the △SNP was used to pot.

### 4.4. RNA-seq

Samples were harvested from *M98* and *D1* plants after the low-temperature treatments for total RNA extraction. The RNA was extracted using the Eastep® Super Total RNA Extraction kit (Promega, Beijing, China), and the RNA integrity was determined using the Agilent 2100 bioanalyzer. Oligo dT magnetic beads were used to enrich mRNA from total RNA. After fragmentation, first-strand cDNA was synthesized using random hexamer primers, followed by the synthesis of second-strand cDNA. The library was prepared after end repair, the addition of an A-tail, ligation of adapters, fragment selection, amplification, and purification. The library used was an Illumina HiSeq Xten sequencing platform for high-throughput sequencing (NoveGene, Nanjing, China). The sequencing read length was PE150, generating more than 6 G of raw data per sample. The original data were filtered to obtain Clean Data using the fastp v0.23.1 software analysis process. Sequence comparisons were performed using the rapeseed reference genome (ZS11.v0), the ZS11.v0 index was constructed using Hisat2 v2.0.5, and paired-end clean reads were aligned to the ZS11.v0 using the same version of Hisat2 (v2.0.5). The mapped data obtained were evaluated for library quality using insertion fragment length and randomness tests. Structure-level analyses, such as variable splicing analysis, new gene discovery, and gene structure optimization, were conducted. Differential expression analysis, functional annotation, and functional enrichment of DEGs were performed according to the expression levels of the genes in different samples or sample groups. Differential expression analysis was performed using the DESeq2 R package v1.20.0, with an FDR of ≤ 0.01 and differential screening multiple thresholds of ≥ 2. Subsequently, GO and KEGG were performed using the clusterProfiler R package v3.8.1.

### 4.5. Time-Course Analysis

MaSigPro (version 1.58.0) [92] was used for time-course analysis. As there were excessive DEGs at seven time points, the DEGs showed the same trend at three consecutive time points as in the gene set. We set the polynomial regression degree to nine and used the forward selection algorithm for stepwise regression, with an R-squared threshold of >0.7, to extract significantly fitting genes as time-course DEGs. hclust was performed based on the correlation distance.

### 4.6. Weighted Gene Co-Expression Network Construction

Co-expression networks were constructed using weighted gene co-expression network construction (WGCNA) (version 1.70) [93]. Time-course DEGs of *D1* (N = 930) and *M98* (N = 919) were used for WGCNA unsigned co-expression network analysis. The modules were obtained using the automatic network construction function, blockwiseModules, with default settings. Among the identified modules, the gray module, representing failed classified genes, was considered invalid. The top 5 genes of the module eigengene-based connectivity (kME) were selected as hub genes. The interaction network between hub genes and their directly associated genes was visualized using Cytoscape (version 3.9.0) [52].

### 4.7. Merging Results from MaSigPro and Weighted Gene Co-Expression Network

The gene clustering through MaSigPro was based on their similar expression profiles, while the algorithm of WGCNA was based on the correlation among the genes. To obtain the common genes selected through these two approaches, we integrated the clustering results of the time-course with the co-expression network modules. The top three percentiles were selected to form new modules.

### 4.8. qRT-PCR

According to BSA-seq and RNA-seq, candidate genes related to the cold response were verified through qRT-PCR. RNAs were extracted from the leaves of rapeseed *M98* and *D1* plants after low-temperature treatment and reverse-transcribed into cDNA as a template for qRT-PCR. FastStart Universal SYBR Green Master Mix (Magen, Guangzhou, China) was used in the QuantStudio 5 PCR System (Thermo Fisher Scientific, Waltham, MA, USA) for qRT-PCR. The procedure was 95 °C for 1 min, 40 amplification cycles (95 °C 10 s, 55–64 °C 10 s, and 72 °C 30 s). After PCR, the temperature was increased from 65 to 95 °C for melting curve detection to determine the specificity of amplification. Each reaction was repeated thrice. Relative expression levels were calculated using the 2^−ΔΔCt^ method, with *BnaACTIN* used as the normalizer for total RNA level normalization in the qRT-PCR assays. The primers used are listed in Appendix A.

## 5. Conclusions

In this study, BSA-seq and RNA-seq were used to identify cold resistance genes. *BnaC09G0354200ZS*, *BnaC09G0353200ZS*, and *BnaC09G0356600ZS* were identified as candidate genes that may lead to the difference in cold resistance between rapeseed *D1* and *M98*, which requires further functional validation. These findings facilitate the exploration of the cold resistance mechanism in rapeseed and offer effective data support and a theoretical basis for the subsequent functional validation of candidate genes, providing significant insights into rapeseed breeding and genomic research.

## Figures and Tables

**Figure 1 ijms-26-01148-f001:**
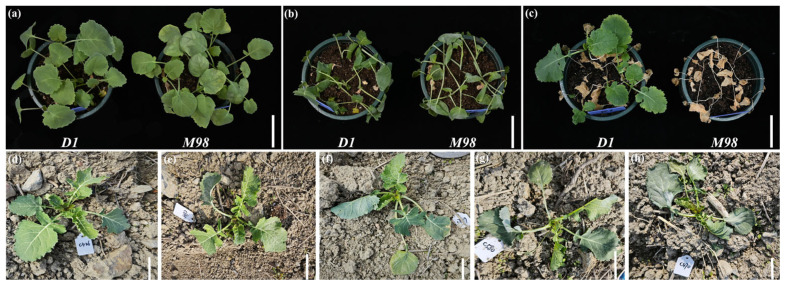
Phenotypic observation of rapeseed seedlings in response to cold stress. (**a**) Rapeseed *D1* and *M98* seedlings grown under normal conditions (22 °C) in the plant chamber. (**b**) Phenotypes of *D1* and *M98* seedlings after cold treatment (−4 °C, 24 h) in the plant chamber. (**c**) *D1* and *M98* seedlings at 24 days of recovery (22 °C) in the plant chamber. The survival rates were evaluated at this stage, as shown in the images. (**d**–**h**) The phenotype of seedlings with cold damage level 0–4 in the field, respectively. Bar, 5 cm (**a**–**h**).

**Figure 2 ijms-26-01148-f002:**
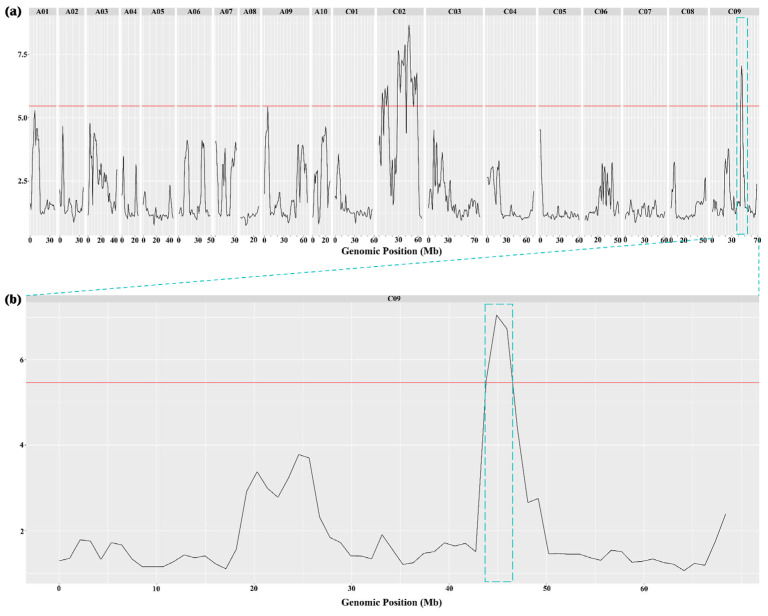
Manhattan plots showing the distribution of G’-Value on the chromosomes. (**a**) Manhattan plots showing the distribution of G’-Value on all 19 chromosomes. The blue box is a single pink on the chromosome C09. (**b**) Enlarged view of chromosome C09 in (**a**), highlighting the single peak on chromosome C09. The blue box is the candidate interval for this study.

**Figure 3 ijms-26-01148-f003:**
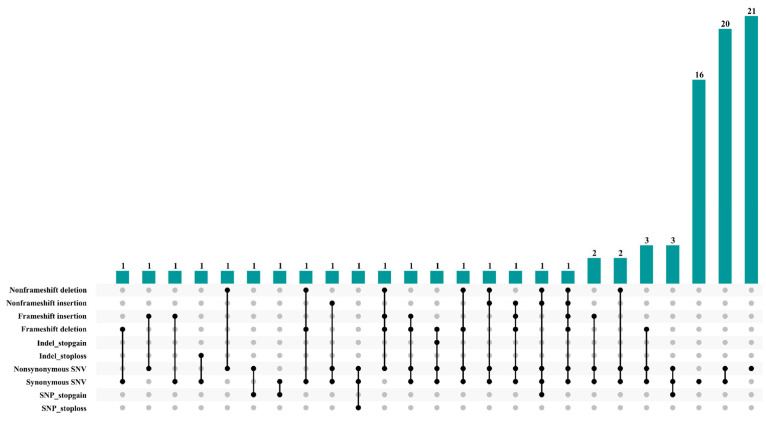
Variation types of candidate genes in the candidate interval (CI).

**Figure 4 ijms-26-01148-f004:**
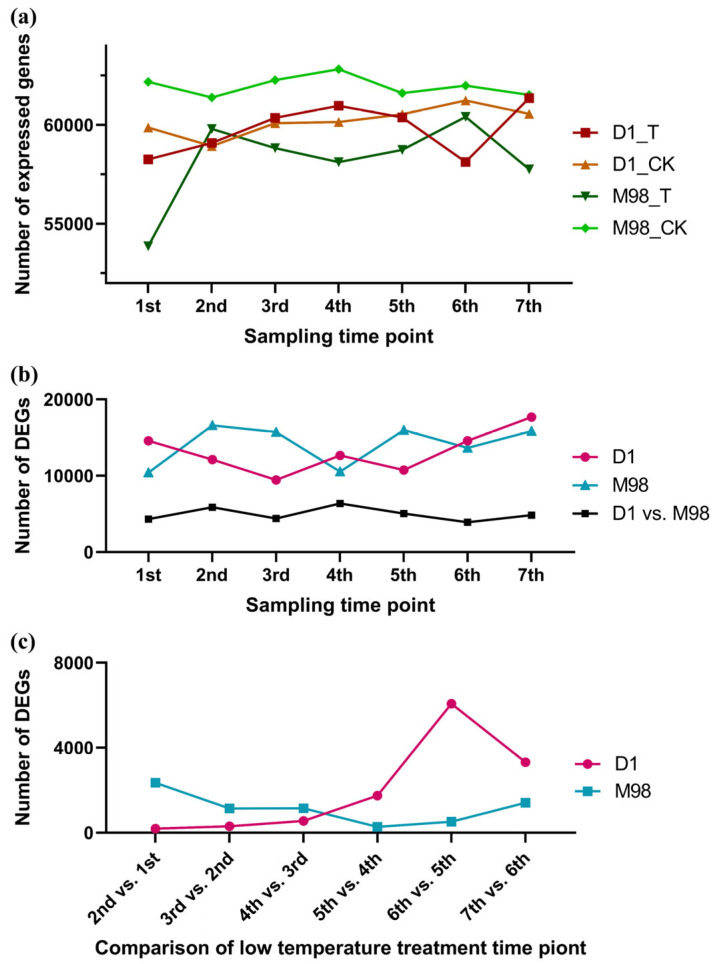
Statistics of the expressed genes. (**a**) Statistics of the expressed genes in all samples. (**b**) Statistics of the DEGs in *D1*, *M98,* and D1 vs. M98. (**c**) Statistics of the DEGs between two adjacent low-temperature treatment time points in *D1* and *M98*. Line charts showing the number of the expressed genes during different sampling time points (1st–7th are described in the methods) in rapeseed *D1* and *M98*. CK: control; T, low-temperature treatment; D1 vs. M98, a gene set after removing D1_CK from the D1_T vs. a gene set after removing M98_CK from the M98_T.

**Figure 5 ijms-26-01148-f005:**
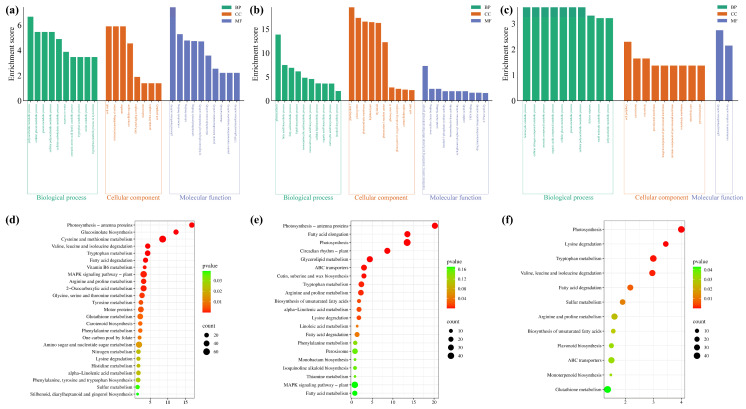
Enrichment analyses of the GO annotation and KEGG pathway. The GO annotation analysis of gene set in the 6th vs. 5th time points in *D1* (**a**) and *M98* (**c**). The GO annotation analysis of the intersection of the 5th and 6th time points in *D1* (**b**). The KEGG pathway analysis of gene set in the 6th vs. 5th time points in *D1* (**d**). The KEGG pathway analysis of the intersection of the 5th and 6th time points in *D1* (**e**) and *M98* (**f**).

**Figure 6 ijms-26-01148-f006:**
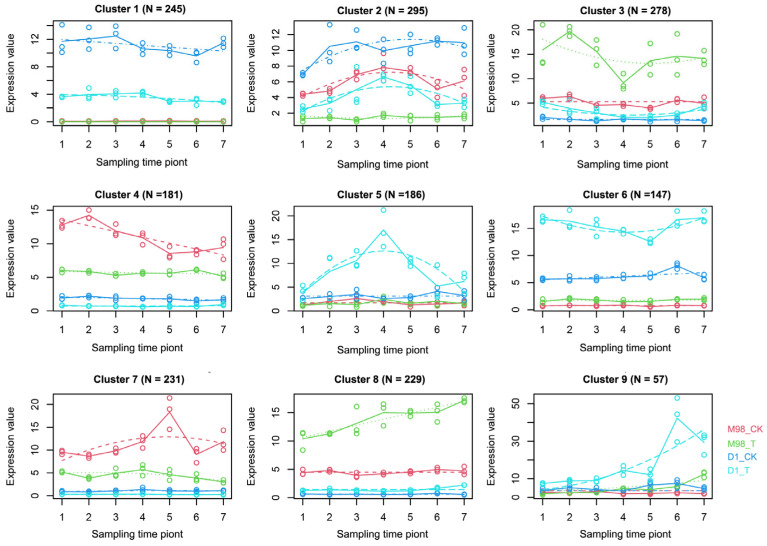
Different gene expression patterns based on the time-course analysis. Each cluster represents a trend of gene expression, and the numbers at the bottom indicate the number of genes in the cluster. Different colored curves represent cultivars under low-temperature treatment conditions, and each curve represents the median profile of genes at different low-temperature treatment time points. CK, control; T, treatment; Sampling time points 1st–7th are described in the methods.

**Figure 7 ijms-26-01148-f007:**
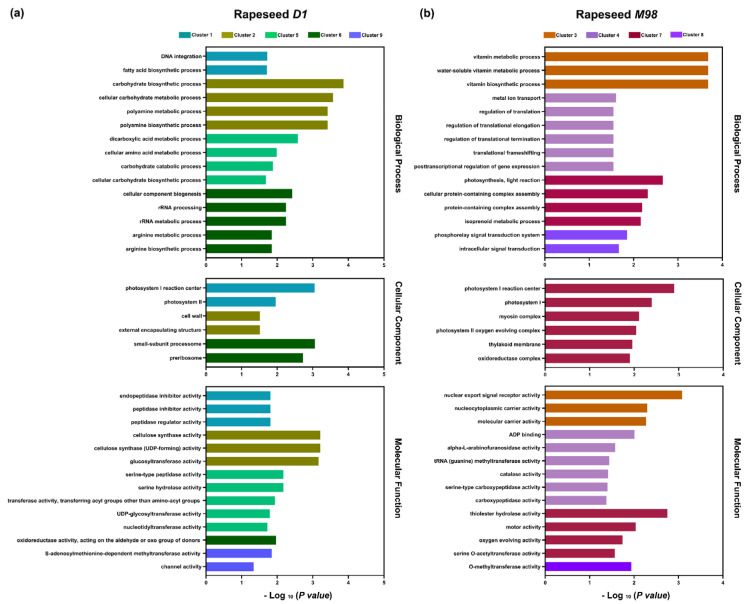
GO enrichment analysis of the predominant expression in *D1* and *M98*. GO enrichment analysis of the predominant expression in *D1* (**a**) and *M98* (**b**), including biological process, cellular component, and molecular function. The different colors represent different clusters. The X-axis represents −log_10_ (*p*-value), and the enriched GO terms are indicated on the Y-axis.

**Figure 8 ijms-26-01148-f008:**
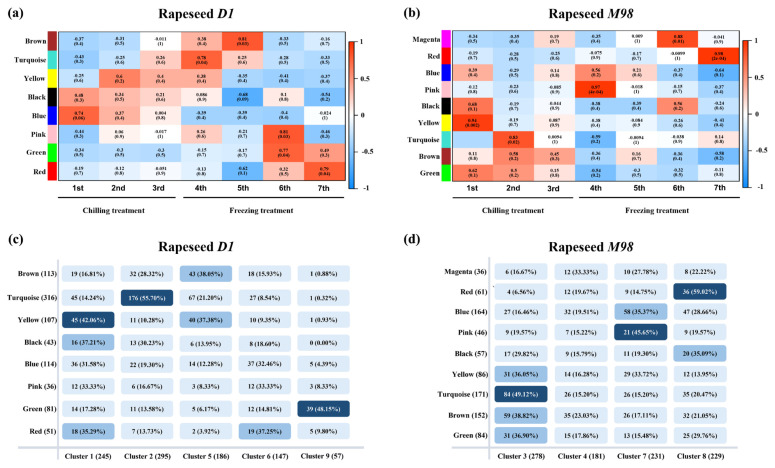
Co-expression network construction and overlapping analysis with MaSigPro. (**a**,**b**) Weighted gene co-expression network analysis of the genes with dominant expression in *D1* and *M98* at seven time points of low-temperature treatment. Each row represents a module, and the correlation coefficient and the *p*-value calculated using Fisher’s exact test are shown in each square. The table is color-coded by correlation according to the color legend. The intensity and direction of correlations are indicated on the right-hand side of the heat map (red, positive; blue, negative). (**c**,**d**) Overlapping analyses of genes in the five clusters and eight modules in *D1* and the four clusters and nine modules in *M98*. Sampling time points 1st–7th are described in the methods.

**Table 1 ijms-26-01148-t001:** Union set of three intersection genes of CI and CR1, CR2, and CR3.

ZS11 Gene ID	Gene_chr	Gene_start	Gene_end	Gene _length	SNP or InDel Location	Arabidopsis Gene ID	Description	Reference
BnaC09G0348300ZS	C09	44269434	44272091	2658	intergenic, upstream, coding region	AT5G52850	Encodes a trichome protein.	Wienkoop et al., 2004 [61]
BnaC09G0348700ZS	C09	44331062	44332915	852	intergenic, upstream, coding region, downstream	AT5G52880	Encodes a F-box family protein.	Kuroda et al., 2012 [62]
BnaC09G0349600ZS	C09	44407051	44409392	771	intergenic, upstream, coding region	AT5G52882	*AT5G52882* is a nitrate responsive gene, encodes a P-loop containing nucleoside triphosphate hydrolases superfamily protein.	Vidal et al., 2013 [63]
BnaC09G0349700ZS	C09	44477743	44481064	1338	intergenic, upstream, coding region	AT5G52890	Encodes an AT hook motif-containing protein.	Araport11
BnaC09G0351400ZS	C09	44707220	44720688	1155	intergenic, upstream, coding region	AT5G53110	RING/U-box superfamily protein	Araport11
BnaC09G0351800ZS	C09	44770576	44771014	297	intergenic, upstream, coding region	/	/	/
BnaC09G0352400ZS	C09	44856686	44857590	555	intergenic, upstream, coding region, downstream	AT5G53160	Encodes RCAR3, a regulatory component of ABA receptor. Interacts with protein phosphatase 2Cs ABI1 and ABI2. Stimulates ABA signaling. Overexpression of PYL8/RCAR3 produces hypersensitivity to ABA in seed germination and increased tolerance to water stress in vegetative tissues.	Saavedra et al., 2010 [64]
BnaC09G0353200ZS	C09	44949935	44951760	432	intergenic, coding region, downstream	AT3G13470	Encodes a chloroplast chaperonin, CPN60B2, which is related to cold-resistant proteins.	Trentmann et al., 2020 [65]
BnaC09G0354200ZS	C09	45117551	45118636	1086	intergenic, upstream, coding region, downstream	AT5G53290	*AT5G53290* is rapidly and strongly induced under low temperatures, encodes a member of the ERF (ethylene response factor) subfamily B-5 of ERF/AP2 transcription factor family. It responded to cold via a cytokinin two-component signaling (TCS)-independent pathway.	Park et al., 2015; Jeon et al., 2016 [66,67]
BnaC09G0356000ZS	C09	45492486	45494637	873	intergenic, upstream, coding region, downstream	AT5G53420	AT5G53420 responses to N-treatment, encodes a CCT motif family protein CCT101.	Li et al., 2020 [68]
BnaC09G0356200ZS	C09	45509221	45512694	2285	intergenic, upstream, coding region	AT5G53450	Encodes FBN11 protein. Responses to osmotic stress.	Choi et al., 2021 [69]
BnaC09G0356600ZS	C09	45562081	45563185	813	intergenic, coding region, downstream	AT5G53490	Encodes thylakoid lumenal 17.4 kDa protein (TL17).	Gollan et al., 2021 [70]
BnaC09G0356700ZS	C09	45563969	45567618	3389	intergenic, upstream, coding region	AT5G53500	Encodes a transducin/WD40 repeat-like superfamily protein CB5E.	Araport11
BnaC09G0357400ZS	C09	45684736	45687186	573	upstream, coding region	AT5G53560	Encodes a cytochrome b5 isoform E (CB5E). When *CB5E* is overexpressed with *FAD2* or *FAD3*, can enhance the capacity of w-6 desaturation or w-3 desaturation.	Kumar et al., 2012 [53]
BnaC09G0359400ZS	C09	45918847	45919488	642	intergenic, coding region, downstream	AT5G53730	Encodes a phloem-specific membrane protein, NHL26. Overexpression of NHL26 alters phloem export and sugar partitioning in *Arabidopsis*.	Vilaine et al., 2013 [71]
BnaC09G0359700ZS	C09	45980345	45983640	1710	intergenic, upstream, coding region	AT5G53760	Encodes mildew resistance locus of protein MLO11. *Atmol11* mutant shown an abnormal root phenotype.	Bidzinski ET AL., 2014 [72]; Nguyen et al., 2016 [73]
BnaC09G0359800ZS	C09	45995055	45998065	1816	intergenic, upstream, coding region	AT5G53770	Encodes a nucleotidyltransferase family protein TRL (TRF4/5-like), which is a terminal nucleotidyltransferase that is mainly responsible for oligoadenylation of rRNA precursors in *Arabidopsis*.	Sikorski et al., 2015 [74]
BnaC09G0360900ZS	C09	46180257	46181292	954	intergenic, coding region	AT5G53870	Encodes early nodulin-like protein 1 ENODL1. *AtENODL1* is a strong candidate gene associated with geotropism.	Wilkinson et al., 2021 [75]
BnaC09G0361200ZS	C09	46225547	46226986	1440	coding region	AT5G53890	Encodes a leucine-rich repeat receptor kinase (LRR-RK) involved in the perception of phytosulfokine (PSK), which is a 5-aa tyrosine-sulfated peptide that primarily promotes cellular proliferation.	Kaufmann et al., 2021 [76]

## Data Availability

The raw data of BSA-seq and RNA-seq were submitted to NCBI with accession numbers PRJNA1208240 and PRJNA1208758, respectively.

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
