# Peer review of "Combined Bulked Segregant Analysis-Sequencing and Transcriptome Analysis to Identify Candidate Genes Associated with Cold Stress in Brassica napus L"

_ijms, 2025, doi:10.3390/ijms26031148_

Round 1

Reviewer 1 Report

Comments and Suggestions for Authors

The authors conducted a study on Combined Bulked Segregant Analysis-Sequencing and Transcriptome Analysis to Identify Candidate Genes Associated with Cold Stress in Brassica napus. The study is rich in content, academically articulated, logically structured, and well-written. However, several issues need to be addressed:

Lines 39–41: This section should be merged into the preceding paragraph as it currently appears isolated.

Line 51: The acronym ICE should be introduced with its full form when mentioned for the first time. Similarly, line 53, COR etc.

Lines 141–143: Figure 1c is referenced before Figure 1a and 1b, which is inconsistent. Additionally, from the description, Figure 1a and 1b could logically be placed before Figure 1c.

Line 140: How is the survival rate (SR) of leaves defined? Does complete drying mean death? There may be partially withered leaves, and the authors should clarify this point.

Line 146: Regarding the 0–4 grading system, the authors should provide relevant images, to make it clearer for the readers.

Line 162: average coverage depth of 41.29%? Is that 41.29×?

Lines 179–185: This content is not closely related to this section. It may be better placed after the discussion of GO or KEGG enrichment in this section for better coherence.

Line 187: “7th interval” is unclear. Furthermore, the authors excluded six intervals on C02 and focused on C09. What was the basis for this decision? Did the authors analyze the intervals on C02?

Line 214: “Low-temperature response is a dynamic and complex process that usually decreases the number of expressed genes,” requires refs to support this claim.

Line 238: “The GO analysis of the gene.” It is unclear whether this refers to GO analysis based on DEGs or something else. This phrasing needs clarification.

Figure S13: Regarding the qPCR validation, how did the authors standardize the data? Was it normalized based on the M98_CK at the 1st time point? Additionally, the relationship between the selected genes and cold tolerance should be explicitly explained.

Lines 342–344: For the WGCNA analysis, “Based on the genes predominantly expressed in D1 (N = 930) and M98 (N = 919).” How was “predominantly expressed” defined? This needs clarification.

Lines 469-474: The authors should not only focus on verifying the reliability of the transcriptome data in this section. Instead, they should perform further analysis based on qPCR, compare the significance differences between groups, and provide proper explanations to deepen the study.

Lines 519-533: This section requires restructuring, especially starting from "Therefore." The CR3 and C1 intersection analysis should be considered an independent analysis, and it does not have a causal relationship with the preceding text. The authors should clarify this point.

Lines 617-618: Please specify the specific software tools used for Clean Data processing to ensure the transparency and reproducibility of the analysis.

Lines 619-623: The specific software tools used for the reference-guided assembly in RNA-seq need to be clearly specified.

Lines 624-627: The specific software tools used for DEG analysis, as well as KEGG and GO analysis, need to be explicitly mentioned.

The discussion requires further depth. For example, how do homologous genes of BnaC09G0354200ZS perform in other species? The study should consider expanding its scope to a broader context. Additionally, further explanations of some results can be moved to the discussion section to enhance its depth. The current discussion primarily relies on the results and lacks sufficient in-depth analysis.

Lines 556-558: “Survival rate (SR), a phenotypic indicator of cold tolerance, was calculated by dividing the number of surviving plants by the total number of plants. Each experiment involved over 30 seedlings per line, and each line had at least three biological replicates.”. The definition of SR here is completely different from the one mentioned earlier: “Rapeseed D1 was determined to be a cold-tolerant line that exhibited a 75% survival rate (SR) with healthy leaves after cold stress.” The authors need to clarify how SR is specifically calculated to avoid confusion for readers.

Did the authors consider the impact of genomic bias in the BSA analysis? Notably, in the BSA results, significant signals were concentrated on the C genome, while the A genome did not reach the threshold. Have the authors considered the potential influence of genomic bias? Furthermore, were similar bias results observed in the RNA-seq analysis?

As an OA journal, to ensure the reproducibility of the study, it is recommended that the authors upload the raw data to a publicly accessible database after the manuscript is accepted and provide the access link in the paper.

Author Response

Comments 1: Lines 39–41: This section should be merged into the preceding paragraph as it currently appears isolated.

Response 1: It has been merged into the previous paragraph (lines 38-41), thanks for your suggestion. Thank you.

Comments 2: Line 51: The acronym ICE should be introduced with its full form when mentioned for the first time. Similarly, line 53, COR etc.

Response 2: The same errors have been corrected, such as those in lines 52-91, 115-121, 406-423 and 517-536. Thank you.

Comments 3: Lines 141–143: Figure 1c is referenced before Figure 1a and 1b, which is inconsistent. Additionally, from the description, Figure 1a and 1b could logically be placed before Figure 1c.

Response 3: The description order in the article has been adjusted to Figure 1a and 1b before 1c (lines 145-149). Thank you.

Comments 4: Line 140: How is the survival rate (SR) of leaves defined? Does complete drying mean death? There may be partially withered leaves, and the authors should clarify this point.

Response 4: The survival rate refers to the ratio of number of plants surviving after cold stress to the total number of plants treated, which has been redescribed in 4.1 (lines 612-613). The surviving seedlings may have partially withered leaves. Thank you.

Comments 5: Line 146: Regarding the 0–4 grading system, the authors should provide relevant images, to make it clearer for the readers.

Response 5: The other cold damage levels have been added to Figure 1 (lines 158-165). Thank you.

Comments 6: Line 162: average coverage depth of 41.29%? Is that 41.29×?

Response 6: It is 41.29×, it has been corrected (line 170). Thank you.

Comments 7: Lines 179–185: This content is not closely related to this section. It may be better placed after the discussion of GO or KEGG enrichment in this section for better coherence.

Response 7: This content has been removed to the Introduction (lines 92-99). Thank you.

Comments 8: Line 187: “7th interval” is unclear. Furthermore, the authors excluded six intervals on C02 and focused on C09. What was the basis for this decision? Did the authors analyze the intervals on C02?

Response 8: Because BSA-seq results showed that seven peak values were significant, but except for a single peak located on chromosome C09, the remaining six peaks were clustered on chromo-some C02. Moreover, the six peaks on chromosome C02 were not only closely clustered but also covered more than half the chromosome, suggesting that this may be due to the background differences between the parents. Based on the current results, we believe that the major cold resistance gene is most likely located on chromosome C09 in this study, so we have prioritized the identification of candidate genes in the individual interval on C09. As for the six large, connected intervals clustered together on chromosome C02, we are further investigating the underlying reasons for their formation. The corresponding changes have been made in the text (lines 187-193). Thank you.

Comments 9: Line 214: “Low-temperature response is a dynamic and complex process that usually decreases the number of expressed genes,” requires refs to support this claim.

Response 9: This sentence is wrong. It has been changed to “Low-temperature response is a dynamic and complex process that usually chang-es the expression pattern of many genes.”, and the references have been added (lines 219-220). Thank you.

Comments 10: Line 238: “The GO analysis of the gene.” It is unclear whether this refers to GO analysis based on DEGs or something else. This phrasing needs clarification.

Response 10: Here, GO analysis was performed using DEGs, and it has been corrected (line 244). Thank you.  

Comments 11: Figure S13: Regarding the qPCR validation, how did the authors standardize the data? Was it normalized based on the M98_CK at the 1st time point? Additionally, the relationship between the selected genes and cold tolerance should be explicitly explained.

Response 11: Yes, the expression level of each gene in M98-CK1 was set as 1. It has been added to the diagram in supplementary S13. The relationship between the candidate genes and cold tolerance were shown in the Discussion (lines 545-571). Thank you.   

Comments 12: Lines 342–344: For the WGCNA analysis, “Based on the genes predominantly expressed in D1 (N = 930) and M98 (N = 919).” How was “predominantly expressed” defined? This needs clarification.

Response 12: Based on the time-course analysis,the dynamic expression pattern was divided into two dominant expression types. The first type was the dominant expression pattern of D1 (N = 930) and was represented by Clusters 1, 2, 5, 6, and 9. The second type was M98 (N = 919), which was represented by Clusters 3, 4, 7, and 8 (Figure 6). The information was described in lines 303-306. And Lines 348-351 has been modified to improve readability. Thank you.

Comments 13: Lines 469-474: The authors should not only focus on verifying the reliability of the transcriptome data in this section. Instead, they should perform further analysis based on qPCR, compare the significance differences between groups, and provide proper explanations to deepen the study.

Response 13: This is a good suggestion, and the corresponding analysis has been added (lines 468-476). Thank you.

Comments 14: Lines 519-533: This section requires restructuring, especially starting from "Therefore." The CR3 and C1 intersection analysis should be considered an independent analysis, and it does not have a causal relationship with the preceding text. The authors should clarify this point.

Response 14: This section has been reorganized to include the analysis of DEGs and their relationship with cold tolerance. It was found that none of the DEGs are located on chromosome C09, and the overlap rate of time-course analysis and WGCNA was low. Therefore, to avoid missing potential cold-resistant genes, all genes in CR3 intersected with all genes in CI (lines 545-570). Thank you.

Comments 15: Lines 617-618: Please specify the specific software tools used for Clean Data processing to ensure the transparency and reproducibility of the analysis.

Response 15: The specific software tool and its version has been added to lines 672-673. Thank you.

Comments 16: Lines 619-623: The specific software tools used for the reference-guided assembly in RNA-seq need to be clearly specified.

Response 16: The specific software tool and its version has been added to lines 674-676. Thank you.

Comments 17: Lines 624-627: The specific software tools used for DEG analysis, as well as KEGG and GO analysis, need to be explicitly mentioned.

Response 17: The specific software tools and their versions have been added to lines 681-684. Thank you.

Comments 18: The discussion requires further depth. For example, how do homologous genes of BnaC09G0354200ZS perform in other species? The study should consider expanding its scope to a broader context. Additionally, further explanations of some results can be moved to the discussion section to enhance its depth. The current discussion primarily relies on the results and lacks sufficient in-depth analysis.

Response 18: The discussion on the functions of DEGs and candidate genes in the plant kingdom has been added, along with an analysis of the relationship between DEGs and candidate genes. It was found that some DEGs are closely associated with candidate genes, and there may also be interactions between the candidate genes themselves (lines 517-539, 545-571 and 577-599). Thank you.

Comments 19: Lines 556-558: “Survival rate (SR), a phenotypic indicator of cold tolerance, was calculated by dividing the number of surviving plants by the total number of plants. Each experiment involved over 30 seedlings per line, and each line had at least three biological replicates.”. The definition of SR here is completely different from the one mentioned earlier: “Rapeseed D1 was determined to be a cold-tolerant line that exhibited a 75% survival rate (SR) with healthy leaves after cold stress.” The authors need to clarify how SR is specifically calculated to avoid confusion for readers.

Response 19: The SR has been redescribed in 4.1 (lines 612-613). Thank you.

Comments 20: Did the authors consider the impact of genomic bias in the BSA analysis? Notably, in the BSA results, significant signals were concentrated on the C genome, while the A genome did not reach the threshold. Have the authors considered the potential influence of genomic bias? Furthermore, were similar bias results observed in the RNA-seq analysis?

Response 20: This is a very good question. We had not noticed it. And there were no similar biased results were observed in the RNA-seq analysis. For example, among the 12 candidate genes in lins399-415, 5 of them are located on the A genome. Thank you.

Comments 21: As an OA journal, to ensure the reproducibility of the study, it is recommended that the authors upload the raw data to a publicly accessible database after the manuscript is accepted and provide the access link in the paper.

Response 21: The raw data of BSA-seq and RNA-seq were submitted to NCBI with accession numbers PRJNA1208240 and PRJNA1208758, respectively. They have been added to the Data Availability Statement (lines 739-740). Thank you.

Reviewer 2 Report

Comments and Suggestions for Authors

Summary
The manuscript offers a valuable evaluation of the candidate genes involved in cold stress response in Brassica napus. For this purpose, authors used a BSA-and RNA-seq combined approach. While the study is insightful and well-executed, several improvements in content and presentation would enhance its clarity and impact.

Title
Please include “L.” after the Latin name Brassica napus for taxonomic accuracy.

Abstract

  • Line 23: Following the enumeration of candidate genes for cold stress response, briefly mention their molecular functions. This will provide readers with a quick understanding of how these genes may contribute to cold resistance.

Introduction

  • The introduction effectively outlines the molecular mechanisms underpinning cold resistance, emphasizing the role of candidate differentially expressed genes (DEGs) and their potential connection to reactive oxygen species (ROS) accumulation.
  • Consider incorporating additional references on transcriptome assembly approaches (reference-based or de novo) for identifying abiotic stress resistance genes in the Brassica genus: https://doi.org/10.1016/j.stress.2024.100657; https://doi.org/10.3389/fgene.2022.958217
  • Lines 119–135: Significantly condense this paragraph. Briefly state the aim of the study while avoiding detailed methodology or results, which belong to their respective sections.

Results

  • Figure 5: Improve the graphical quality of the KEGG pathway plot. A higher-resolution image will ensure clarity.
  • Enhance visualization by providing a network plot that links DEGs (with their fold changes) to the KEGG pathways. This will allow readers to better grasp the affected molecular pathways under cold stress conditions.
  • Table 1: Add more details about the listed genes, such as chromosomal localization or the amino acid chain length of the encoded proteins/enzymes.
  • If applicable, predict the effects of missense mutations identified in the BSA-seq data. Tools such as AlphaFold (AlphaFold Server), ChimeraX, or PyMOL can be utilized to model and visualize these mutations. This approach would add a compelling layer to the analysis.

Discussion

  • Expand the discussion to include a deeper analysis of molecular pathways potentially affected by the regulation of candidate genes.
  • Use tools such as STRING or Cytoscape to identify and visualize common pathways that are altered, integrating these insights with your findings.

Materials and Methods
This section is informative and well-detailed, requiring no major revisions.

Author Response

Comments 1: Please include “L.” after the Latin name Brassica napus for taxonomic accuracy.

Response 1: “L.” has been added. Thank you.

Comments 2: Line 23: Following the enumeration of candidate genes for cold stress response, briefly mention their molecular functions. This will provide readers with a quick understanding of how these genes may contribute to cold resistance.

Response 2: Due to the 200-word limit for the abstract in IJMS, the molecular functions of the four candidate genes cannot be included. Readers interested in these genes can refer to the full article for further details. Thank you.

Comments 3: The introduction effectively outlines the molecular mechanisms underpinning cold resistance, emphasizing the role of candidate differentially expressed genes (DEGs) and their potential connection to reactive oxygen species (ROS) accumulation.

Response 3: The functional analysis of DEGs and candidate genes in the plant kingdom has been included in the discussion. Upon analysis, it was found that they are not closely related to ROS, so no content related to ROS has been added (lines 517-539 and 545-571). Thank you.

Comments 4: Consider incorporating additional references on transcriptome assembly approaches (reference-based or de novo) for identifying abiotic stress resistance genes in the Brassica genus: https://doi.org/10.1016/j.stress.2024.100657; https://doi.org/10.3389/fgene.2022.958217

Response 4: It is a good suggestion. They were added to the text in lines 131-132. Thank you.

Comments 5: Lines 119–135: Significantly condense this paragraph. Briefly state the aim of the study while avoiding detailed methodology or results, which belong to their respective sections.

Response 5: This paragraph has been condensed (lines 134-141). Thank you.

Comments 6: Figure 5: Improve the graphical quality of the KEGG pathway plot. A higher-resolution image will ensure clarity.

Response 6: Figure 5 has been improved the graphical quality (line 291). Thank you.

Comments 7: Enhance visualization by providing a network plot that links DEGs (with their fold changes) to the KEGG pathways. This will allow readers to better grasp the affected molecular pathways under cold stress conditions.

Response 7: The purpose of the KEGG analysis in Figure 5 is to demonstrate that there are many cold tolerance-related pathways during the 6th vs. 5th period, and the changes in this period are closely related to the cold-tolerant D1 line and the cold-sensitive M98 line. We analyzed whether we could link DEGs to KEGG pathways but found that only one gene identified by Time-course and WGCNA appeared once, and none of the three candidate genes were involved. Therefore, we believe there is no need to modify the figure to avoid interfering with the main focus. We sincerely appreciate the reviewer’s suggestion.

Comments 8: Table 1: Add more details about the listed genes, such as chromosomal localization or the amino acid chain length of the encoded proteins/enzymes.

Response 8: Some details have been added to the Table 1 (line 433). Thank you.

Comments 9: If applicable, predict the effects of missense mutations identified in the BSA-seq data. Tools such as AlphaFold (AlphaFold Server), ChimeraX, or PyMOL can be utilized to model and visualize these mutations. This approach would add a compelling layer to the analysis.

Response 9: We attempted to use AlphaFold, but currently, the regulatory pathways of our candidate genes in the entire plant kingdom are unclear, and the upstream and downstream target genes, as well as interacting proteins, are all unknown. Therefore, this aspect will not be presented in the paper for now. Thank you for your suggestion. Thank you.

Comments 10: Expand the discussion to include a deeper analysis of molecular pathways potentially affected by the regulation of candidate genes.

Response 10: The combined analysis of DEGs and candidate genes has been conducted, and it was found that some genes are closely related to the candidate genes, which has been added to the discussion. In addition, we also discovered that there may be interactions between candidate genes, which could jointly regulate the cold tolerance process in rapeseed. Thank you.

Comments 11: Use tools such as STRING or Cytoscape to identify and visualize common pathways that are altered, integrating these insights with your findings.

Response 11: Due to the unclear regulatory pathways of the candidate genes, we are unable to visualize the altered pathways. We sincerely appreciate the valuable suggestions from you.

Comments 12: This section is informative and well-detailed, requiring no major revisions.

Response 12: The SR has been redescribed in 4.1 (lines 612-613). The specific software tools in this study and their versions have been added to lines 672-673, 674-676 and 681-684. Thank you.

Round 2

Reviewer 2 Report

Comments and Suggestions for Authors

Authors addressed all the Reviewer's comments